# How big five personality traits influence information sharing on social media: A meta analysis

Hao Lin[1,2,3], Chundong Wang[1,2,3], Yongjie Sun[4]*

1 School of Computer Science and Engineering, Tianjin University of Technology, Tianjin, China, 2 Tianjin Key Laboratory of Intelligence Computing and Novel Software Technology, Tianjin University of Technology, Tianjin, China, 3 Engineering Research Center of Learning-Based Intelligent System, Ministry of Education, Tianjin, China, 4 School of Languages and Culture, Tianjin University of Technology, Tianjin, China

* sunny1970-2-9@163.com

**Data Availability Statement:** All relevant data are within the manuscript and its Supporting information files.

**Funding:** This work was supported by the National Natural Science Foundation of China Joint Fund

## Abstract

Research interest in information sharing behavior on social media has significantly increased over the past decade. However, empirical studies on the relationship between Big Five personality traits and information sharing behavior have yielded contradictory conclusions. We aimed to investigate how Big Five personality influences information sharing behavior on social media. This meta-analysis systematically reviewed high-quality studies indexed by web of science and CNKI from the past decade ($n = 27$, with 31969 samples) and performed a meta-analysis to examine the association between Big Five personality traits and information sharing behavior. The literature search was performed in September 2023. The meta-analysis results showed that extraversion ($\beta = 0.05^{**}$) had a positive relationship with information sharing behavior on social media. Agreeableness ($\beta = -0.06^{**}$), conscientiousness ($\beta = -0.03^{**}$), and neuroticism ($\beta = -0.03^{**}$) had negative relationships with information sharing behavior on social media. However, the relationship between openness and information sharing behavior was not clearly observed due to insufficient research. The meta-analysis results are made available to the scientific community to enhance research, comprehension, and utilization of social media.

## 1 Introduction

Social users continuously expand the scale of their presence as they engage in acquiring, sharing, and interacting with information, thereby maintaining, strengthening, or reconstructing their existing social relationships [1, 2]. Research on the constant dissemination of diverse information on social media can contribute to detecting rumors [3, 4], improving recommendation systems [5], marketing [6], managing social media [7, 8] and employee [9], and addressing other related areas.

The sharing behavior of social media users largely determines the dissemination of information on social media [10]. Personality traits, which are stable patterns of thoughts, feelings, and behaviors, have a significant influence on human cognitive patterns [11–13]. This makes

Project [U1536122], Key Special Project of Technology Boosts Economy 2020 by Ministry of Science and Technology [SQ2020YFF0413781], Pilot Demonstration Project of Big Data Industry Development [Big data intelligent analysis and service platform for language barrier regeneration applications], and Tian Jin Research Innovation Project for Postgraduate Students [2022BKY158].

**Competing interests:** NO authors have competing interests.

it an excellent starting point for studying information sharing behavior on social media. The Big Five personality model is the most commonly used personality taxonomy in information science [14]. The Big Five personality traits are often integrated into research methods in areas such as rumors, recommendation systems, employee management, etc. However, although many review studies have investigated the correlation between the Big Five personality traits and human behavior [11, 15–17], the association between personality and information sharing behavior has not been thoroughly examined. In addition, there is no published meta-analysis on the relationship between two. The conclusions drawn between the two remain complex and unclear. This may lead to introducing personality traits in the method, which could be counterproductive. So, in this meta-analysis, we identified the connection between Big Five personality traits and information sharing behavior through a meta-analysis, which is a quantitative literature review method, to more effectively examine the human element in information science. The results may point to one of the bottlenecks faced by personalized research in information science.

This paper begins by providing definitions of Big Five personality traits and Information sharing behavior. Then, the research questions and hypotheses of this study are proposed. The adopted materials and methods of the meta-analysis are subsequently described. Section 5 presents the results of the meta-analysis, which is subsequently followed by a discussion of these results in section 6.

## 2 Theoretical review

### 2.1 Big five personality

Personality has been defined as "psychological qualities that contribute to an individual's enduring and distinctive patterns of thinking, feeling, and behaving." Various theorists have developed several models of personality, each representing different perspectives, such as Five Factor Model, Myers–Briggs Type Indicator, Eysenck's three factor model, and seven-factor personality model.

The Five Factor Model, also known as the "Big Five" model of personality, is the most widely accepted and well-known theory within the dispositional perspective of personality. Big Five personality comprises five broad traits: extraversion (EXT), agreeableness (AGR), conscientiousness (CON), neuroticism (NEU) (or called emotional stability), and openness (OPN). The chaotic nature and lack of easy conclusions can be observed in agreeableness, conscientiousness, and openness and social media behavior. For instance, according to Indu et al. [18], individuals characterized by high extroversion and low agreeableness tend not to disseminate rumors. However, Buchanan [19] proposes that those who propagate false information feature lower agreeableness, yet exhibit higher levels of extroversion and neuroticism. As another example, various literature report the relationship between openness and information sharing behavior. However, contrary to these reports, a study by Zuniga et al. [20] found no significant correlation between these variables in a large sample. Meta-analysis is suitable for summarizing these confusing conclusions.

### 2.2 Information sharing behavior on social media

Due to its real-time, interactive, and diverse content characteristics, social media has gradually become an essential aspect of people's daily lives over time. On these platforms, various types of information are shared, including entertainment information, health information, emergency information, political information, even rumor [18] and fake news [21]. The continuous sharing of such information enhances the appeal and usefulness of social media. In social

media, information publishing, commenting, and forwarding all belong to information sharing behaviors, which transmit useful information to others.

Multiple psychometric measurement instruments have been created to evaluate individuals' willingness to share information, including the Knowledge-sharing Behaviours Scale [22] and Information Exchange Scale [23]. However, no recognized specific scale has been developed to assess willingness to share information specifically on social media platforms. In addition to designing questionnaires, currently, researchers evaluate this willingness primarily through statistical design media data and interviews.

# 3 Research questions and hypothesis

The purpose of this meta-analysis was to consolidate all existing empirical evidence on the connection between Big Five personality traits and information sharing behavior on social media. So, **RQ1**:Do the information sharing behavior of users on social media correlate with their (a) extraversion, (b) agreeableness, (c) conscientiousness, (d) neuroticism, and (e) openness?

The extraversion trait distinguishes social, proactive individuals who are oriented towards themselves from silent, serious, shy, and quiet individuals. It is often considered highly positively correlated with the frequent use of social media [17, 24, 25]. Neuroticism reflects the individual emotional regulation process. Those with high neuroticism exhibit heightened reactivity to external stimuli compared to the general population, and they generally struggle with regulating and responding to emotions, often experiencing negative emotional states. It is often considered negatively correlated with the sharing behavior, since individual with high neuroticism tend to feel shy, anxious, insecure, and awkward in social situations [26, 27]. Agreeableness measures an individual's attitude towards others. Conscientiousness distinguishes between individuals who are trustworthy and meticulous from those who are lazy and careless. Openness refers to an individual's cognitive style, their capacity to tolerate unfamiliar situations, and their aptitude for exploration. So, based on our explanation of the Big Five personality traits and the previous literature of personality traits and human cognitive abilities [11], the following hypotheses were proposed for the meta-analysis:

- **H1**: Extraversion is positively related to information sharing behavior on social media.

- **H2**: Agreeableness is negatively related to information sharing behavior on social media.

- **H3**: Conscientiousness is positively related to information sharing behavior on social media.

- **H4**: Neuroticism is negatively related to information sharing behavior on social media.

- **H5**: Openness is positively related to information sharing behavior on social media.

# 4 Material and methods

## 4.1 Literature searching and screening

A thorough literature search was conducted by two independent researchers in September 2023, among major databases involving Web of Science® and China National Knowledge Infrastructure® (CNKI). To ensure the quality of the literature, we only selected literatures indexed by Science Citation Index or Engineering Index (for literature written in English) and Core Journals of Peking University (http://hxqk.lib.pku.edu.cn/) (for literature written in Chinese). This rule filters out most low-quality articles in CNKI [28]. Due to the timeliness of

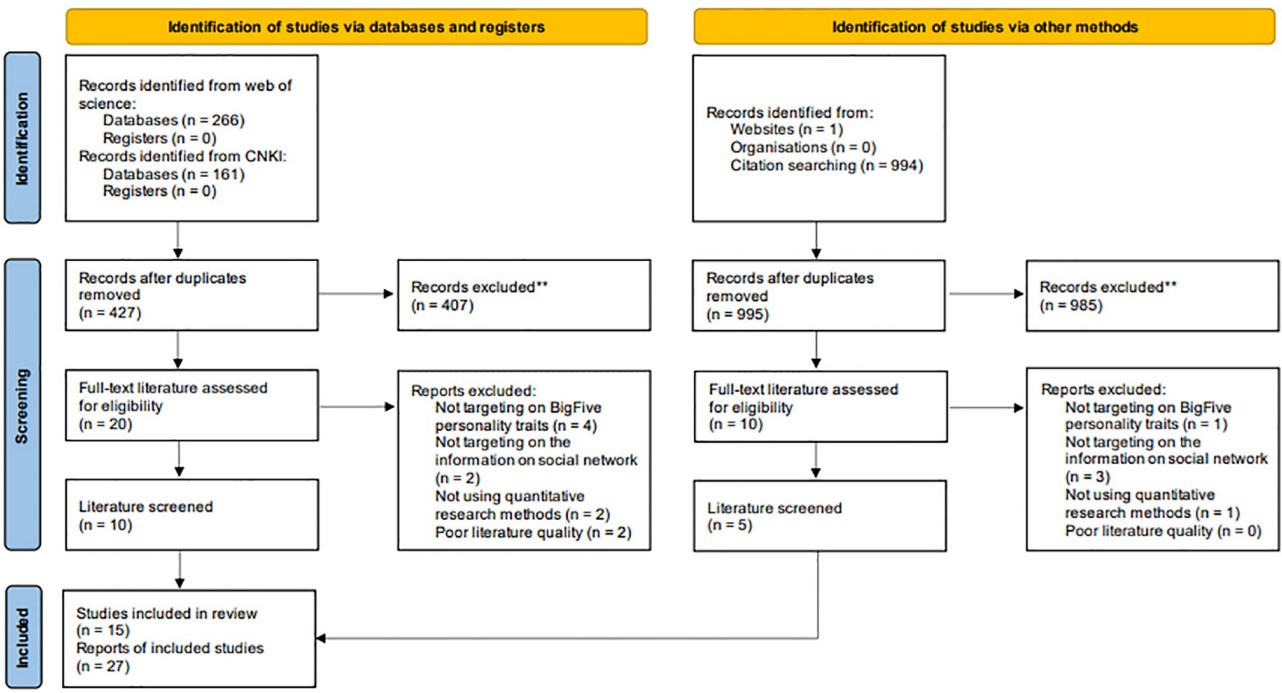

**Fig 1. PRISMA flow diagram of the meta-analysis.** A paper may present findings from numerous studies.

social media research, we only adopt literature from the past decade (2012–2023). Given the main research objectives, we searched literature with multiple keywords, which include "personality", "social media", "information", "sharing", "dissemination", "public opinion", "人格" "传播", "社交媒体" "舆情" and "信息共享" by Boolean search terms. We also gathered references from literature identified for inclusion in the meta-analysis, thereby adding five more papers to the final meta-analysis.

In addition to screening for journal papers and conference papers, relevant dissertations (e. g., [29]) were also screened to avoid potential bias. We subsequently screened the eligible papers using one criterion. That is, the paper must adopt Big Five as the personality taxonomy (e. g., [30]), and it's research object must be the share behavior of social media. After eliminating duplicated articles resulting from the utilization of multiple databases and channels, the titles and abstracts of all retrieved papers were initially screened, which resulted in a total of 1421 papers at the preliminary stage, and a total of 15 papers and 27 studies in final. The seleection process was also conducted by two independent researchers. The searching and seleection process is shown in Fig 1.

## 4.2 Effect size

Consistent with previous meta-analysis in personality [31–33], our study has applied the standardized regression coefficient $\beta$ and standard error $SE$ as the primary metric to estimate effect size. Almost all paper report $\beta$ between personality traits and information sharing behavior. The larger $|\beta|$, the greater the impact of the independent variable on the dependent variable. For papers reporting other indicators for measuring the degree of statistical distribution (i.e., $t$ in t-distribution, $P$ value, standard deviation $SD$, mean difference $MD$), we use the following

formula to approximately convert these indicators to *SE*.

$$SE = SD/\sqrt{n}. \tag{1}$$

$$SE = MD/t. \tag{2}$$

Where, *n* is total number of samples. *t* can be obtained by consulting the t-value distribution table with the *P* value and the degree of freedom ($n - 1$). When the paper does not provide the precise *P* value but instead presents the level of significance, we proceed with the following approximating.

$$P = \begin{cases} Meaningless & , P = NS \\ 0.05 & , P < 0.05 \\ 0.01 & , P < 0.01 \\ 0.001 & , P < 0.001 \end{cases} \tag{3}$$

## 4.3 Coding and data analysis

According to a predefined coding schema, every paper was coded for the following information: (1) the relevant bibliographic information including the author(s), year of publication, and the country where sample were collected was recorded; (2) the sample characteristics including the number of sample, mean age of the sample, and the percentage of males; (3) the Big Five personality scale used in the study (e.g., BFI, NEO-PI-R, MINI-IPIP, etc.); (4) the information Sharing Scale used in the study (e.g., Likert-type Scale); (5) the effect size of Big Five personality traits assessed in the study.

The random-effects model called DerSimonian-Laird [34] was used to determine whether the average correlations were statistically significant, considering the variation among the included studies. We adopted the *Q* statistic, *H* statistic, and $I^2$ statistic to test the heterogeneity across effect sizes in our chosen papers. The *Q* statistic measures the difference between observed effect sizes and the estimated effect size. The *H* statistic is the correction of the freedom degree for *Q*. The $I^2$ statistic represents the percentage of variability in effect sizes that is unrelated to sampling error. Forest plots visually depict the heterogeneity included in our meta-analysis. In addition, due to the research differences between the Big Five personality traits, we divided the literature into five subgroups for analysis based on the Big Five personality traits. This helps to reduce heterogeneity among the included studies.

## 4.4 Publication bias

Recognizing the publication bias toward positive findings in the personality research community [17], we conducted two methods to determine if any publication bias exists. Firstly, a funnel plot was utilized to visually assess if there were any missing studies with small effect sizes. Next, Begg's test with non parametric rank correlation and Egger's test was employed to provide statistical evidence of publication bias [35].

Overall, Tables 1 and 2 presented the information of all studies included in this meta-analysis. All calculations related to this meta-analysis were conducted in Stata 17.

Fig 2 visually displays the significant publication bias present in our chosen papers. We employed the Leave-one-out method for sensitivity analysis to remove the most extreme outlier studies [36, 37].

**Table 1. Studies included in the meta-analysis.**

| No. | Study | Country | Year | Object | sample | Mean age | % of males | Personality Scale | Information Sharing Scale |
|---|---|---|---|---|---|---|---|---|---|
| 1 | David et al. 2012 (1) [40] | World | 2012 | Social networking service | 300 | 27 | 31 | BFI-44 | Likert-type (1∼7) |
| 2 | David et al. 2012 (2) [40] | World | 2012 | Social networking servic | 300 | 27 | 31 | BFI-44 | Likert-type (1∼7) |
| 3 | Chen 2016 [27] | World | 2016 | Fake message | 171 | 24 | 42.69 | BFI-44 | Likert-type (1∼7) |
| 4 | Liu et al. 2017 [41] | China | 2017 | Social business information | 267 | 22.04 | 44.94 | TIPI-C | Likert-type (1∼7) |
| 5 | Homero et al. 2017 [20] | 20 country | 2017 | Message | 21314 | x | x | NEO-PI-R | Likert-type (1∼10) |
| 6 | Deng et al. 2017 [42] | China | 2017 | Message | 311 | 21.96 | 42.1 | NEO-PI | Likert-type (1∼5) |
| 7 | Mohammad et al. 2018 [25] | World | 2018 | Government information | 257 | 38.91 | 62.01 | NEO-PI-R | Likert-type (1∼7) |
| 8 | Damien et al. 2019 (1) [43] | World | 2019 | Message with emotional information | 197 | 44.9 | 48.94 | BFI-10 | Binary questioning |
| 9 | Damien et al. 2019 (2) [43] | World | 2019 | Message with facial expressions | 197 | 44.9 | 48.94 | BFI-10 | Binary questioning |
| 10 | Buchanan et al. 2019 [44] | USA, UK | 2019 | Fake message | 409 | x | 31.5 | IPIP | Statistics on Facebook |
| 11 | Huang et al. 2020 (1) [45] | China | 2020 | Entertainment information | 317 | 30.26 | 50.16 | Modified BFI | Likert-type (1∼5) |
| 12 | Huang et al. 2020 (2) [45] | China | 2020 | Social Communication Information | 317 | 30.26 | 50.16 | Modified BFI | Likert-type (1∼5) |
| 13 | Huang et al. 2020 (3) [45] | China | 2020 | Self-expression information | 317 | 30.26 | 50.16 | Modified BFI | Likert-type (1∼5) |
| 14 | Yin et al. 2020 [46] | China | 2020 | Negative news | 215 | 22.45 | 41.4 | BFI-44 | Intention of RNI |
| 15 | Tom 2020 (1) [19] | UK | 2020 | Later known fake message | 672 | 44.92 | 47.2 | 41-item BFI | Likert-type (0∼11) |
| 16 | Tom 2020 (2) [19] | UK | 2020 | Later known fake message | 674 | 38.95 | 46.3 | 41-item BFI | Likert-type (0∼11) |
| 17 | Tom 2020 (3) [19] | UK | 2020 | Later known fake message | 650 | 33.07 | 34.6 | 41-item BFI | Likert-type (0∼11) |
| 18 | Tom 2020 (4) [19] | USA | 2020 | Later known fake message | 638 | 44.91 | 44.4 | 41-item BFI | Likert-type (0∼11) |
| 19 | Tom 2020 (5) [19] | UK | 2020 | Known fake message | 672 | 44.92 | 47.2 | 41-item BFI | Likert-type (0∼11) |
| 20 | Tom 2020 (6) [19] | UK | 2020 | Known fake message | 674 | 38.95 | 46.3 | 41-item BFI | Likert-type (0∼11) |
| 21 | Tom 2020 (7) [19] | UK | 2020 | Known fake message | 650 | 33.07 | 34.6 | 41-item BFI | Likert-type (0∼11) |
| 22 | Tom 2020 (8) [19] | USA | 2020 | Known fake message | 638 | 44.91 | 44.4 | 41-item BFI | Likert-type (0∼11) |
| 23 | Xiao et al. 2021 (1) [21] | World | 2021 | Fake news published by social media | 551 | 20.26 | 29.65 | MINI-IPIP | Likert-type (0∼6) |
| 24 | Xiao et al. 2021 (2) [21] | World | 2021 | Fake news published by news media | 551 | 20.26 | 29.65 | MINI-IPIP | Likert-type (0∼6) |
| 25 | Brinda et al. 2022 [47] | India | 2022 | News | 221 | 28.59 | 42 | NEO-PI | Likert-type (1∼5) |
| 26 | Ahmed1 et al. 2022 [48] | Singapore | 2022 | News of COVID-19 | 500 | >21 | x | BFI-44 | Likert-type (1∼5) |
| 27 | Xu et al. 2023 [49] | China | 2023 | Online public opinion on newly emerging infectious diseases | 300 | 20–35 | 44 | Random forest | Statistics on WeiBo |

| No. | Study | Sample | Effect size | | | | | SE | | | | |
|---|---|---|---|---|---|---|---|---|---|---|---|---|
| | | | EXT | AGR | CON | NEU | OPE | EXT | AGR | CON | NEU | OPE |
| 1 | David et al. 2012 (1) [44] | 300 | x | x | -0.248 | x | 0.201 | x | x | 0.194 | x | 0.2 |
| 2 | David et al. 2012 (2) [40] | 300 | x | x | x | 0.119 | x | x | x | x | 0.109 | x |
| 3 | Chen 2016 [27] | 171 | -0.05 | 0.01 | -0.02 | -0.19 | 0.14 | 0.18 | 0.21 | 0.19 | 0.18 | 0.2 |
| 4 | Liu et al. 2017 [41] | 267 | 0.12 | -0.24 | -0.19 | -0.19 | 0.12 | 0.046 | 0.092 | 0.073 | 0.073 | 0.061 |
| 5 | Homero et al. 2017 [20] | 21314 | 0.142 | 0.073 | 0.062 | -0.066 | -0.017 | 0.043 | 0.022 | 0.018 | 0.02 | 0.008 |
| 6 | Deng et al. 2017 [42] | 311 | ∼ | 0.077 | 0.164 | ∼ | ∼ | x | 0.046 | 0.063 | x | x |
| 7 | Mohammad et al. 2018 [25] | 257 | 0.436 | ∼ | ∼ | ∼ | ∼ | 0.13 | ∼ | ∼ | ∼ | ∼ |

*(Continued)*

| 8 | Damien et al. 2019 (1) [43] | 197 | 0.02 | 0.02 | 0.05 | -0.04 | 0.05 | 0.007 | 0.008 | 0.018 | 0.009 | 0.022 |
| 9 | Damien et al. 2019 (2) [43] | 197 | -0.03 | 0.01 | 0.09 | -0.08 | 0.1 | 0.014 | 0.004 | 0.022 | 0.034 | 0.033 |
| 10 | Buchanan et al. 2019 [44] | 409 | 0.07 | -0.15 | -0.07 | 0.04 | -0.02 | 0.03 | 0.05 | 0.03 | 0.03 | 0.04 |
| 11 | Huang et al. 2020 (1) [45] | 317 | 1.738 | -0.112 | -0.318 | -0.273 | -0.27 | 0.184 | 0.143 | 0.141 | 0.097 | 0.173 |
| 12 | Huang et al. 2020 (2) [45] | 317 | 0.84 | 0.275 | 0.164 | -0.208 | -0.13 | 0.127 | 0.116 | 0.114 | 0.076 | 0.137 |
| 13 | Huang et al. 2020 (3) [45] | 317 | 1.191 | -0.234 | 0.298 | -0.318 | -0.192 | 0.144 | 0.173 | 0.119 | 0.08 | 0.144 |
| 14 | Yin et al. 2020 [46] | 215 | -0.184 | 0.15 | 0.151 | -0.041 | 0.038 | 0.086 | 0.065 | 0.069 | 0.074 | 0.066 |
| 15 | Tom 2020 (1) [19] | 672 | 0.036 | -0.085 | -0.053 | -0.007 | -0.04 | 0.023 | 0.03 | 0.024 | 0.023 | 0.027 |
| 16 | Tom 2020 (2) [19] | 674 | 0.04 | 0.005 | 0.001 | 0.027 | 0.039 | 0.018 | 0.031 | 0.02 | 0.02 | 0.024 |
| 17 | Tom 2020 (3) [19] | 650 | 0.057 | 0.004 | -0.05 | 0.036 | 0.02 | 0.019 | 0.03 | 0.022 | 0.022 | 0.025 |
| 18 | Tom 2020 (4) [19] | 638 | 0.02 | -0.048 | -0.067 | -0.005 | 0.012 | 0.017 | 0.028 | 0.021 | 0.02 | 0.022 |
| 19 | Tom 2020 (5) [19] | 672 | 0.054 | -0.173 | -0.043 | -0.001 | 0.013 | 0.032 | 0.039 | 0.031 | 0.03 | 0.037 |
| 20 | Tom 2020 (6) [19] | 674 | 0.034 | -0.049 | 0.006 | 0.065 | 0.03 | 0.026 | 0.044 | 0.03 | 0.029 | 0.036 |
| 21 | Tom 2020 (7) [19] | 650 | 0.042 | -0.054 | 0.003 | 0.075 | 0.042 | 0.042 | 0.035 | 0.027 | 0.028 | 0.032 |
| 22 | Tom 2020 (8) [19] | 638 | 0.045 | -0.106 | -0.027 | -0.008 | -0.024 | 0.02 | 0.033 | 0.025 | 0.024 | 0.026 |
| 23 | Xiao et al. 2021 (1) [21] | 551 | 0.12 | -0.02 | 0.01 | 0.02 | -0.11 | 0.08 | 0.1 | 0.09 | 0.1 | 0.1 |
| 24 | Xiao et al. 2021 (2) [21] | 551 | 0.07 | -0.09 | 0.03 | -0.02 | 0.02 | 0.06 | 0.08 | 0.06 | 0.08 | 0.07 |
| 25 | Brinda et al. 2022 [47] | 221 | 0.192 | -0.002 | -0.202 | 0.195 | 0.291 | 0.081 | 0.1 | 0.09 | 0.087 | 0.12 |
| 26 | Ahmed1 et al. 2022 [48] | 500 | 0.094 | x | -0.167 | x | x | 0.032 | x | 0.045 | x | x |
| 27 | Xu et al. 2023 [49] | 300 | x | x | x | x | x | x | x | x | x | x |

× represents that the effect size cannot be calculated due to insufficient data. ∼ represents that the effect size cannot be calculated due to the personality trait has not been studied.

## 4.5 Quality assessment

Although most quality checklists published in extant academic literature have primarily addressed medical studies, we sought to ensure the thorough evaluation of the selected studies by adhering to a combination of established guidelines. To rigorously assess the methodological quality of the studies included in this meta-analysis, we followed the guidelines of Kitchenham and Charters [38] as well as the meta-analysis on the existing quality assessment tools that are being used in meta-analysis in the area of Engineering [39]. The study suggested using a set of questions based on widely used checklists and guidelines for the design, conduct, analysis, and conclusions of each study in this meta-analysis. The study evaluation criteria were based on the questions presented below.

- **Q1**: Are the aims of the research clearly defined?

- **Q2**: Is there an adequate description of the context in which the research was carried out?

- **Q3**: Was the research design appropriate to address the aims of the research?

- **Q4**: Was there a control group?

- **Q5**: Are the data collection methods adequately described?

- **Q6**: Were all measures used in the study fully defined?

- **Q7**: Is the experimental design appropriate and justifiable?

- **Q8**: Does the study provide description and justification of the data analysis approaches?

- **Q9**: Are the findings of the study clearly stated?

**Table 2. Result of quality assessment.**

| Study | Title | Score |
|---|---|---|
| David et al. 2012 | A tale of two sites: Twitter vs. Facebook and the personality predictors of social media usage | 9 |
| Chen 2016 | The Influences of Personality and Motivation on the Sharing of Misinformation on Social Media | 9 |
| Homero et al. 2017 | Personality Traits and Social Media Use in 20 Countries: How Personality Relates to Frequency of Social Media Use Social Media News Use, and Social Media Use for Social Interaction | 10 |
| Deng et al. 2017 | How do personality traits shape information-sharing behaviour in social media? Exploring the mediating effect of generalized trust | 9 |
| Liu et al. 2017 | 社会化商务下个体心理因素对信息共享行为的影响——大五人格的调节作用 | 9 |
| Mohammad et al. 2018 | Sharing Political Content in Online Social Media: A Planned and Unplanned Behaviour Approach | 10 |
| Buchanan et al. 2019 | Spreading Disinformation on Facebook: Do Trust in Message Source, Risk Propensity, or Personality Affect the Organic Reach of "Fake News"? | 10 |
| Damien et al. 2019 | Willingness to Share Emotion Information on Social Media: Influence of Personality and Social Context | 8 |
| Huang et al. 2020 | 自媒体用户信息共享行为动机分析与实证 | 9 |
| Tom 2020 | Why do people spread false information online? The effects of message and viewer characteristics on self-reported likelihood of sharing social media disinformation | 10 |
| Yin et al. 2020 | Reposting negative information on microblogs: Do personality traits matter? | 9 |
| Xiao et al. 2021 | Wired to seek, comment and share? Examining the relationship between personality, news consumption and misinformation engagement | 9 |
| Brinda et al. 2022 | Fake or real news? Understanding the gratifications and personality traits of individuals sharing fake news on social media platforms | 10 |
| Ahmed1 et al. 2022 | Social Media News use and covid-19 misinformation engagement: Survey study | 9 |
| Xu et al. 2023 | EID事件情境下情绪对信息分享行为的动态影响——人格特质的调节作用 | 8 |
| Kim et al. 2014 | Individual Differences in Social Media Use for Information Seeking | 6 |
| Luo 2018 | 社交媒体中用户人格特质对科学信息分享动机的影响与反思 | 7 |

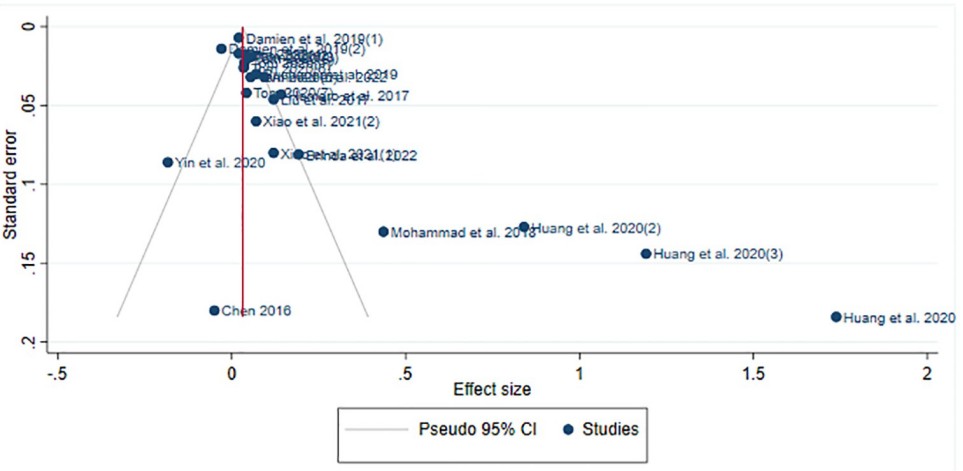

**Fig 2. Funnel plot of our chosen papers (for EXT trait).** The highly asymmetric nature indicates a strong presence of publication bias. Funnel plots for other personality traits are similar to this.

- **Q10**: Does the study add value to academia or practice?

The scoring procedure assigned a value of 1 for "Yes" and 0 for "No". Studies could score between 0 and 10 points. Papers receiving a score exceeding 8 (>8) were decided to be retained in this meta-analysis. The results of quality assessment are presented in Table 2.

## 5 Results

### 5.1 Description of the study

As shown in Tables 1 and 2, the studies were conducted over the world (29.6%), seven studies (25.9%) were conducted in China, and six studies (22.2%) were conducted in UK. In 14 studies (51.8%) social media users were recruited (mean age from 30 to 45 years), in 9 studies (33.3%), the sample comprised social media users (mean age from 20 to 30 years). And the mean age of the sample in 4 studies (14.8%) could not be accurately determined. Most of studies (96.3%) have recruited less than 1000 participants. The following Big Five Personality Scale were adopted: BFI-44 [50] in 4 papers (26.7%), NEO-PI [51] in 2 papers (13.3%), NEO-PI-R [52] in 2 papers (13.3%), 41-item BFI [53] in 2 papers (13.3%) and 9 studies, BFI-10 [54] in 1 paper (6.67%), Mini-IPIP [55] in 1 paper (6.67%), TIPI [56] in 1 paper (6.67%), and machine learning-based method in 1 paper (6.67%). The Big Five personality traits were studied in the 22 studies (81.5%). Likert-type scale were designed to assess users' willingness of information sharing in 22 studies (81.5%). And statistical data on social media were adopted to assess the willingness in 2 studies (13.3%). There were no outliers (i.e., no studies with a SE exceeding 0.21).

### 5.2 Results of meta-analysis

**EXT-information sharing.**    Fig 3 shows the results of DerSimonian-Laird model in EXT subgroup. The DerSimonian-Laird model yield a significant moderate effect size ($\beta = 0.05$, $p(\beta) < 0.001$). The 95% confidence interval (CI) ranged from 0.03 to 0.07. The effect of this subgroup is homogenous ($I^2 = 31.6\% < 50\%$, $H = 1.2 < 1.5$, $Q(16) = 25.28$, $p(Q) = 0.07 > 0.05$).

**AGR-information sharing.**    Fig 4 shows the results of DerSimonian-Laird model in AGR subgroup. The DerSimonian-Laird model yield a significant moderate effect size ($\beta = -0.06$, $p(\beta) < 0.001$). The 95% CI ranged from -0.09 to -0.03. The effect of this subgroup is homogenous ($I^2 = 27.44\% < 50\%$, $H = 1.17 < 1.5$, $Q(13) = 19.59$, $p(Q) = 0.11 > 0.05$).

**CON-information sharing.**    Fig 5 shows the results of DerSimonian-Laird model in CON subgroup. The DerSimonian-Laird model yield a significant small effect size ($\beta = -0.03$, $p(\beta) < 0.001$). The 95% CI ranged from -0.05 to -0.02. This subgroup has no heterogeneity ($I^2 = 9.33\% < 25\%$, $H = 1.04 < 1.2$, $Q(11) = 13.28$, $p(Q) = 0.28 > 0.1$).

**NEU-information sharing.**    Fig 6 shows the results of DerSimonian-Laird model in NEU subgroup. The DerSimonian-Laird model yield a significant small effect size ($\beta = -0.03$, $p(\beta) < 0.001$). The 95% CI ranged from -0.05 to -0.02. This subgroup has no heterogeneity ($I^2 = 21.35\% < 25\%$, $H = 1.12 < 1.2$, $Q(11) = 16.34$, $p(Q) = 0.13 > 0.1$).

**OPN-information sharing.**    Fig 7 shows the results of DerSimonian-Laird model in OPN subgroup. The DerSimonian-Laird model yield a insignificant minor effect size ($\beta = 0.01$, $p(\beta) = 0.3 > 0.01$). The 95% CI ranged from -0.01 to 0.03. Specifically, the CI contains 0, indicating that the relationship between OPN and information sharing is not significant. The effect of this subgroup is homogenous ($I^2 = 29.79\% < 50\%$, $H = 1.19 < 1.5$, $Q(15) = 25.05$, $p(Q) = 0.053 > 0.05$).

Additionally, we conducted a cumulative meta-analysis that sorted by year in EXT, AGR, CON, NEU subgroups, and the results are presented in Fig 8. Obviously, the CIs of each subgroup converge cumulatively.

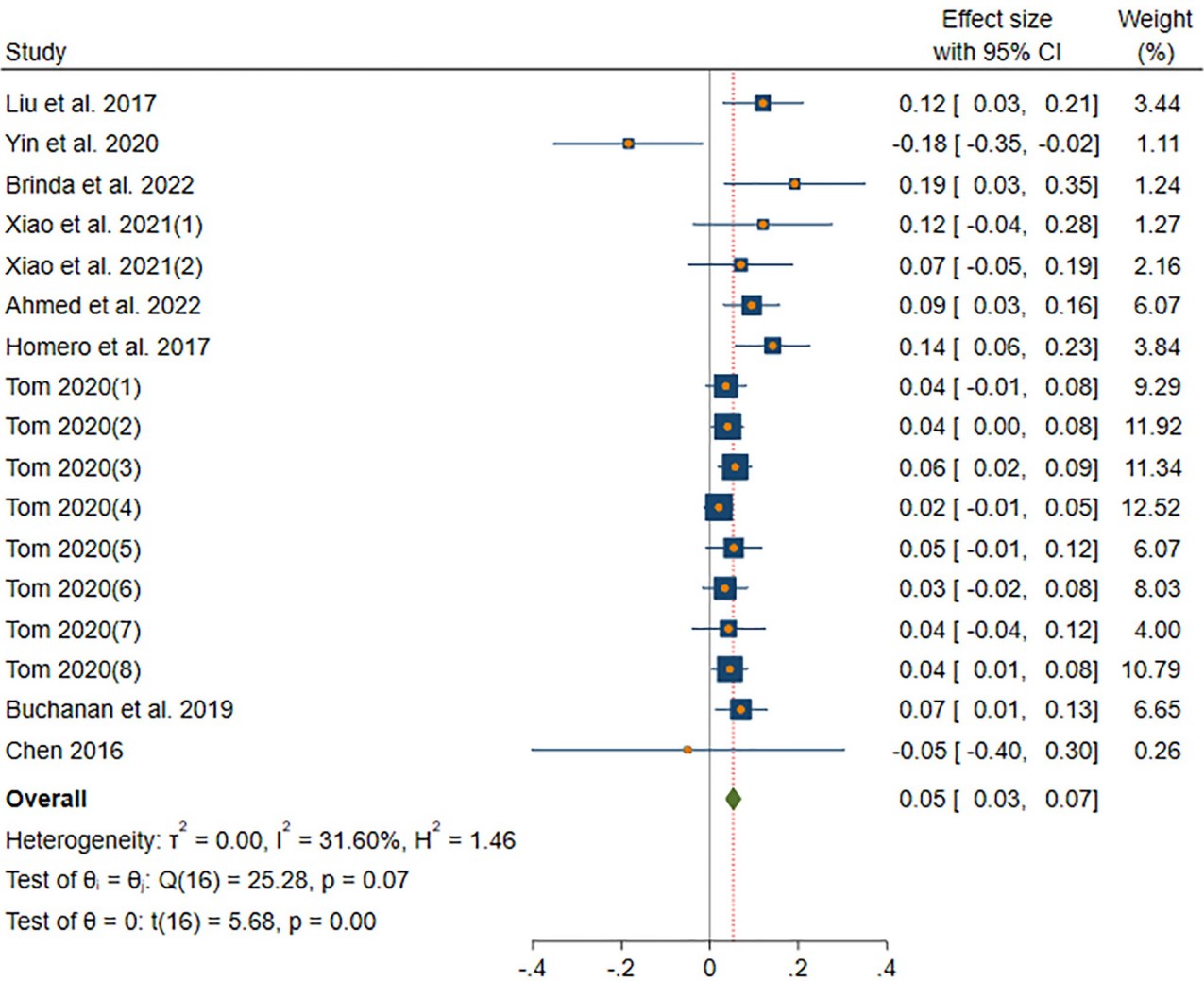

**Fig 3. Forest plot displays the average correlation between the EXT trait and information sharing behavior across multiple studies.**

## 5.3 Publication bias of subgroup analysis

First, the funnel plot (Fig 9) illustrated that the majority of the studies analyzed are evenly distributed in a symmetrical pattern near the center, suggesting no publication bias in our screened studies by Leave-one-out method.

Futher, Table 3 presents the test results of Begg's test and Egger's test, providing the same conclusion as the above. Neither the Begg's test ($p = 0.19, 0.74, 0.788, 0.41, 0.65 > 0.05$) nor the Egger's test ($p = 0.4347, 0.4161, 0.4372, 0.3632, 0.3020 > 0.05$) was signiffcant, providing additional evidence to support the absence of publication bias. This also indicates that there is no publication bias in the studies we selected.

## 6 Discussion

In the long run of research, most of the Big Five personality traits are believed to be related to information sharing behavior on social media. The positive correlation between EXT traits and information sharing behavior is the highest ($\beta = 0.05$). The negative correlation between

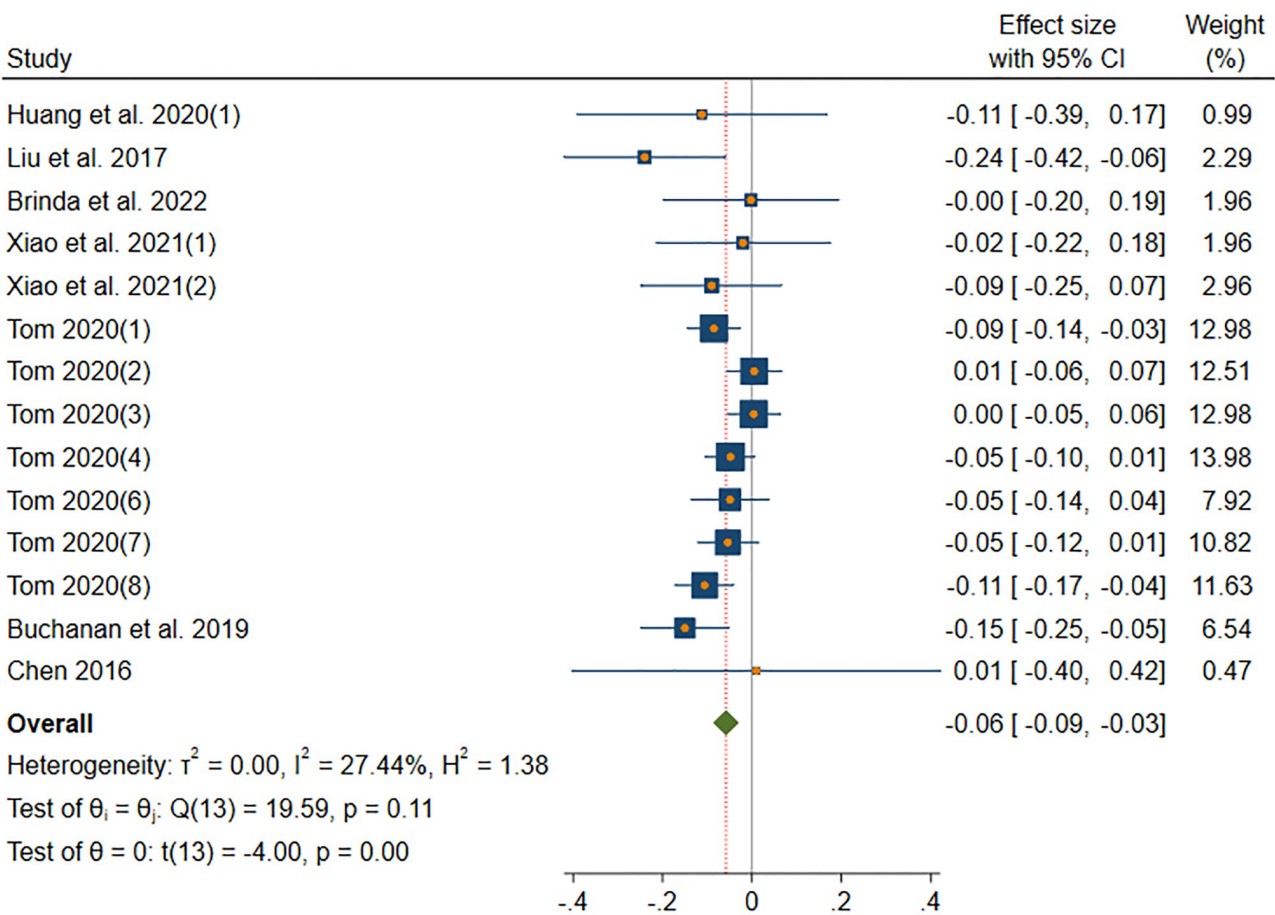

**Fig 4. Forest plot displays the average correlation between the AGR trait and information sharing behavior across multiple studies.**

AGR traits and information sharing behavior is the greatest ($\beta$ = -0.06). Figs 3–6 report significant correlations, therefore, H1, H2, and H4 were supported, H3 was rejected, and H5 was not fully supported. As shown in the above forest plots, literature [20] ($n$ = 21314) and literature [19] ($n$ = 409) have a relatively high weight, since the large sample size. Significantly, no study carries enough weight to decisively influence the outcome, suggesting that our meta-analysis has low sensitivity. Overall, our findings are shown in the Fig 10.

Our research has established a linkage between information sharing behavior on social media and the Big Five personality traits. This evidence implies that future studies related to information sharing behavior, irrespective of their specific scenarios, should incorporate an additional focus on the influence of personality traits. This incorporation will provide a holistic understanding of information sharing behavior. Moreover, studying how personality influences information sharing behavior across different subfields is essential, given the diverse levels of interest that individuals have in various types of information. For example, individuals with high scores of conscientiousness are more likely to participate in discussions on political related information [57].

Second, the key research in current information science is personalized applications [58–60], such as recommendation system and chat AI. The results of this study will contribute to the development of these personalized applications. This work also holds important

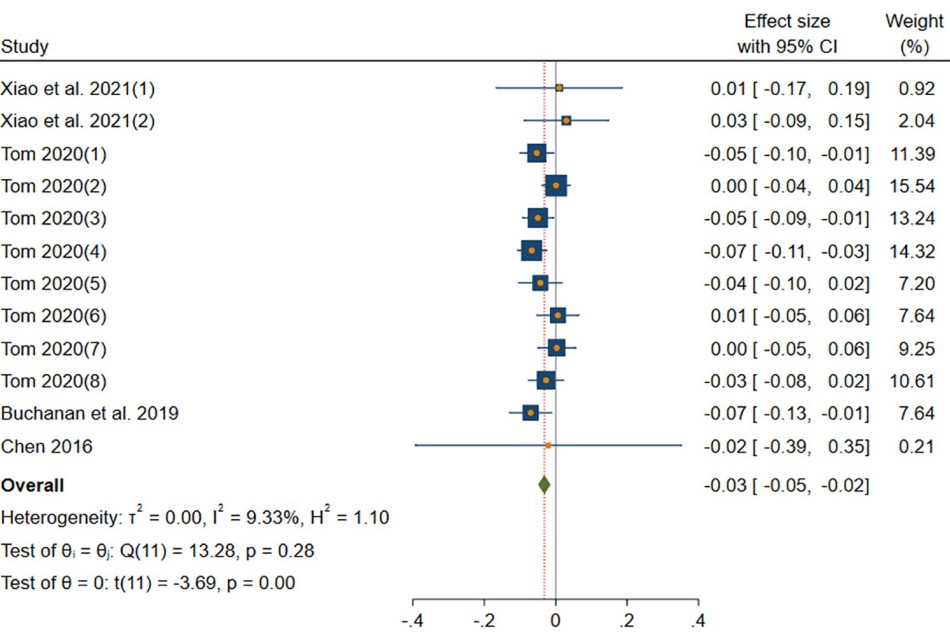

**Fig 5. Forest plot displays the average correlation between the CON trait and information sharing behavior across multiple studies.**

implications for the field of security, particularly in addressing the prevalent issues of rumor spreading and online fraud. Current social landscape is marred by the substantial impact of these problems. The notion of "psychological persuasion" has gained attention in recent research [61], revealing the potency of personalized warnings in improving the efficacy of

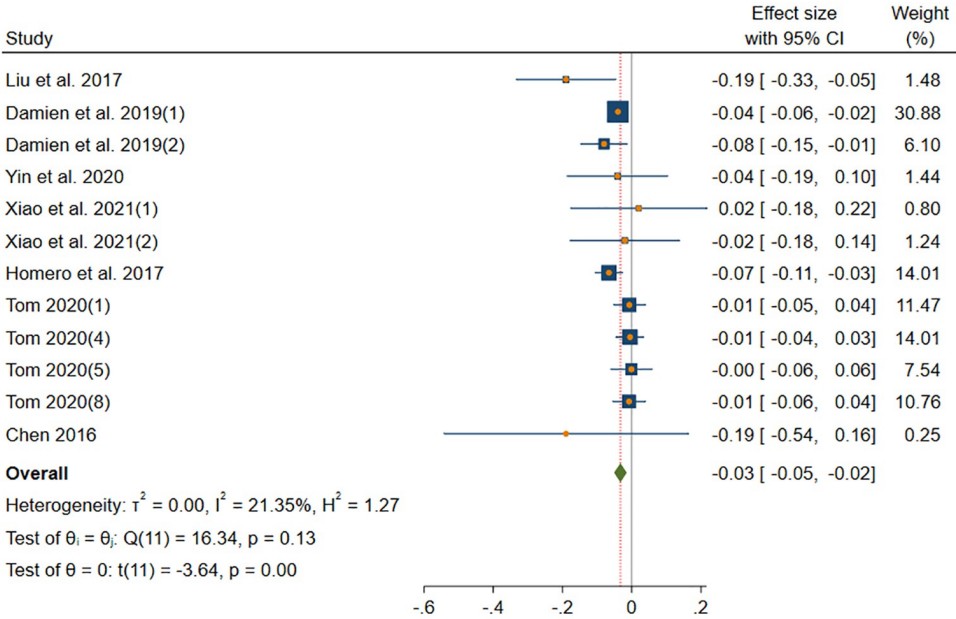

**Fig 6. Forest plot displays the average correlation between the NEU trait and information sharing behavior across multiple studies.**

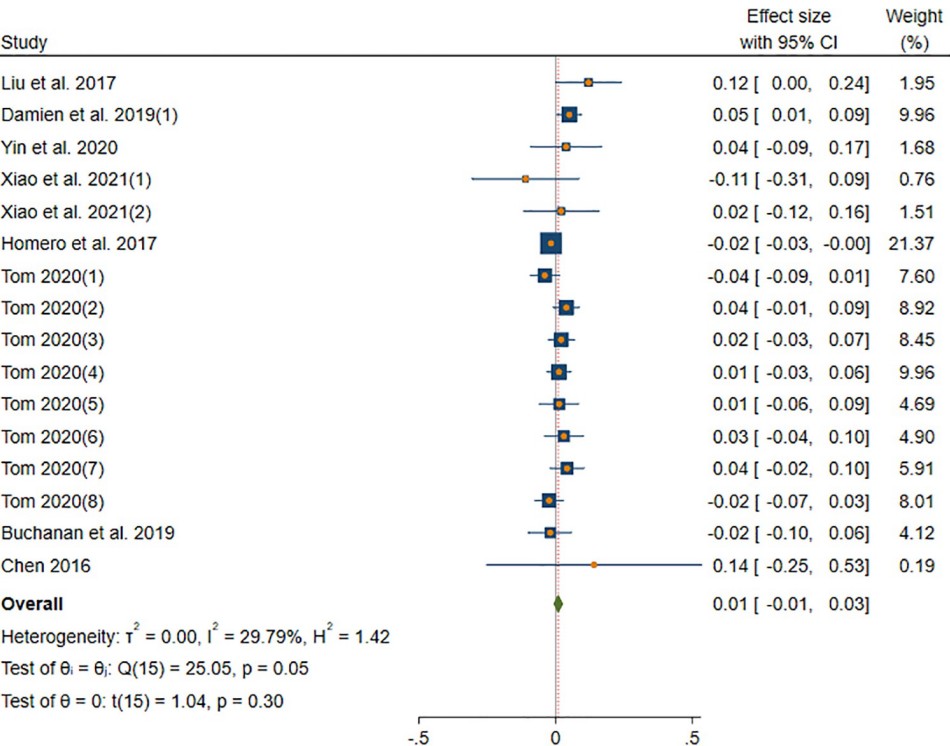

**Fig 7. Forest plot displays the average correlation between the OPN trait and information sharing behavior across multiple studies.**

persuasion strategies [62, 63]. This study, along with machine learning-based automatic personality detection methods, enables the possibility of delivering personalized warnings on a large scale. Finally, the present study's findings are particularly noteworthy in light of the current era of artificial intelligence-generated content (e.g., ChatGPT [64]). The fine-tuning of the large language model may also be based on the user's personality traits. We also recommend personalized strategies when dealing with the dissemination of these generated content messages based on personality traits. Finally, we will discuss the limitations of this work.

## Included studies

Some studies were excluded from this meta-analysis because they did not provide correlation coefficients or regression coefficients. Consequently, the number of included studies was reduced, potentially leading to biased meta-analysis results.

## Cultural differences

Cultural differences will affect the test results of the Big Five personality traits [65, 66]. For example, Europeans and Americans tend to have higher EXT scores compared to Asians and Africans. Unfortunately, inadequate research poses a hindrance to performing subgroup analysis. Insufficient research can result in significant publication bias in meta-analysis.

## Uncertain impact of openness

H5 was not fully supported. This meta-analysis further identified the uncertain impact of openness traits on information sharing behavior. At present, we were not well examined with

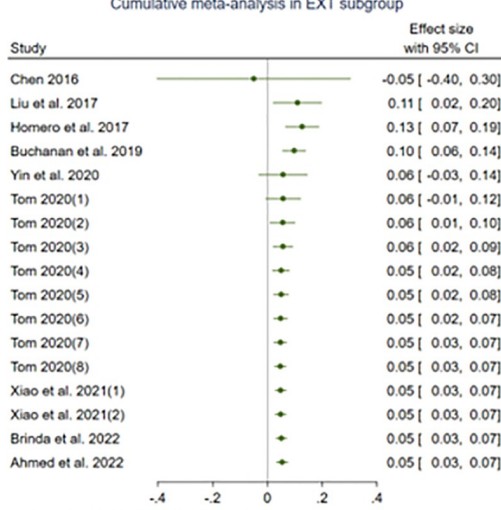

(a) For EXT subgroup.

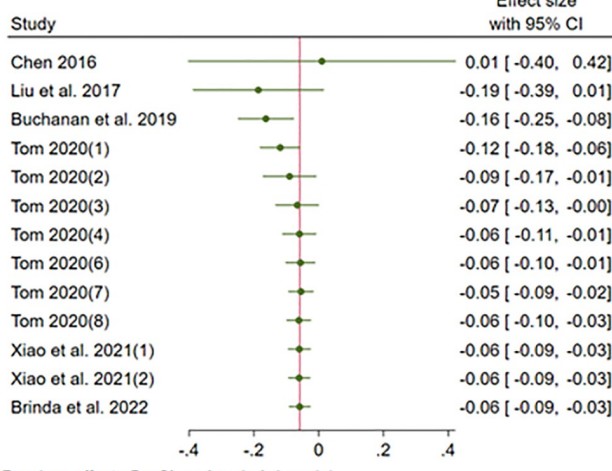

(b) For AGR subgroup.

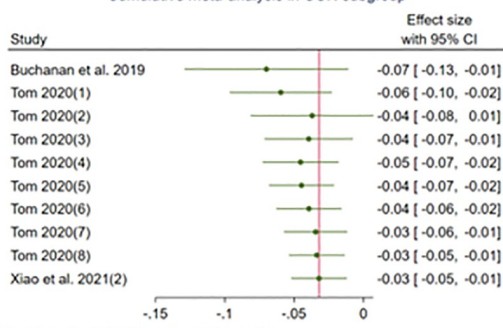

(c) For CON subgroup.

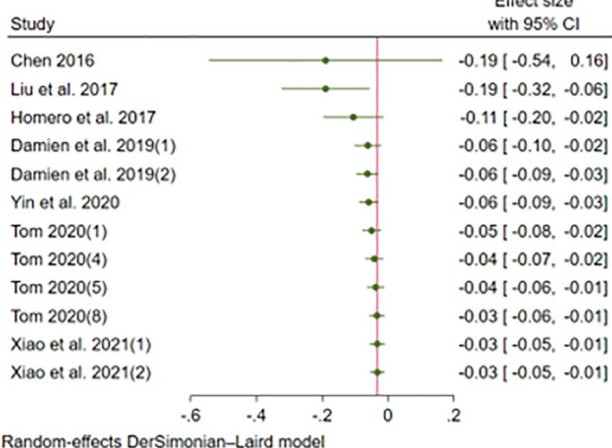

(d) For NEU subgroup.

**Fig 8. Cumulative forest plot.**

enough studies to pass the test (Fig 7). To enhance the validation of these findings in future reviews, more studies published in other languages should be included, along with representative sampling methods.

## Machine learning-based personality measurement

Applying machine learning for social user personality detection allows for a substantial increase in research sample size [67]. However, only one paper has utilized this technology [49], and the level of detection accuracy is concerning.

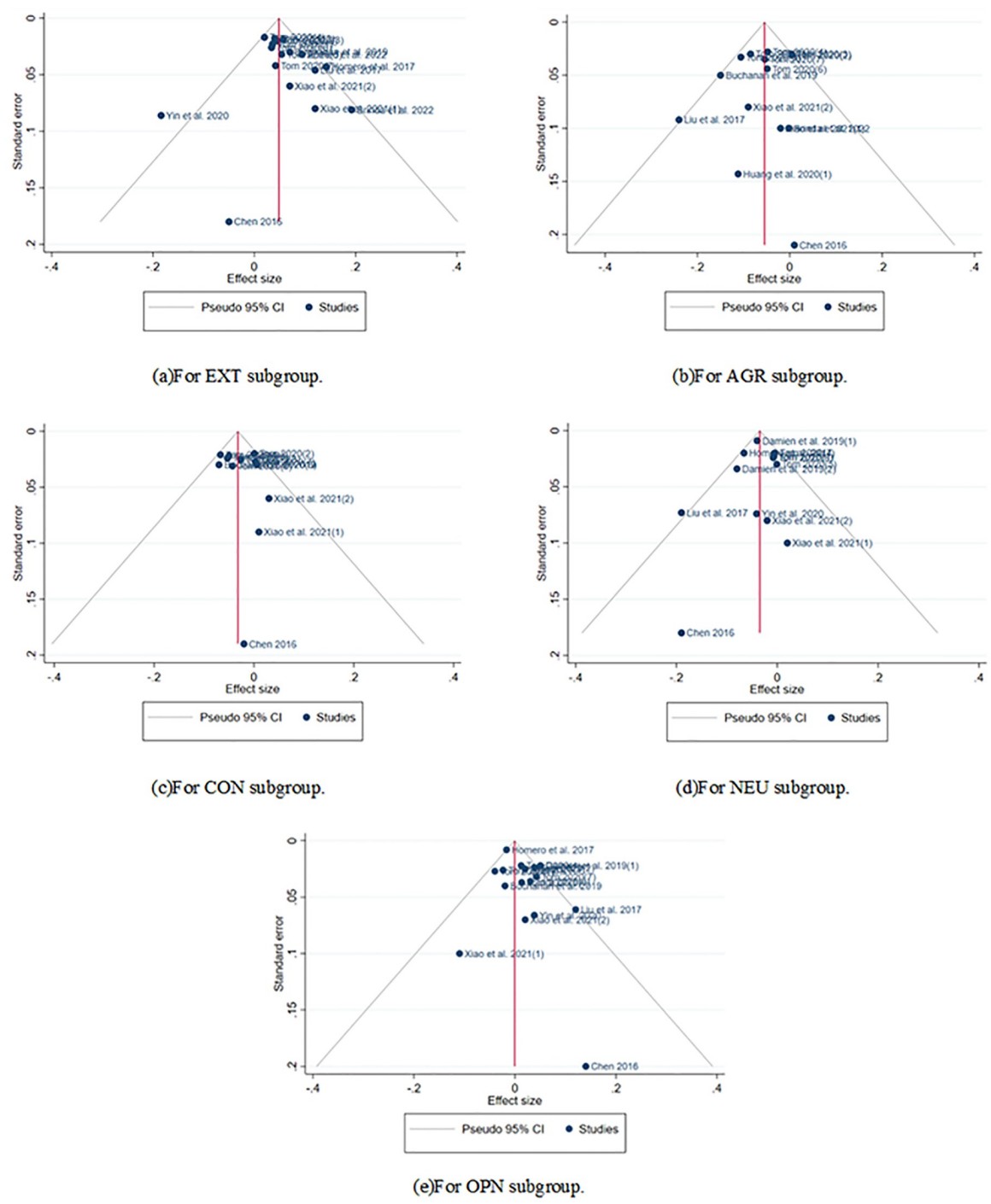

**Fig 9. Funnel plot of OPN subgroup analysis.**

## 7 Conclusion

Although limited, this meta-analysis enhances understanding of the role of personality factors in information sharing behavior on social media in the existing studies. Based on the meta-analysis presented, we found that extraversion positively correlates with information sharing

**Table 3. Begg's test and Egger's test results of subgroup analysis.**

| Subgroup | Sample | Begg's test | | Egger's test | |
|---|---|---|---|---|---|
| | | z | p | z | p |
| EXT | 17 | 1.32 | 0.1871 | 0.78 | 0.4347 |
| AGR | 14 | -0.44 | 0.7418 | -0.81 | 0.4161 |
| CON | 12 | 0.27 | 0.7834 | 0.78 | 0.4372 |
| NEU | 12 | -0.96 | 0.4095 | -0.91 | 0.3632 |
| OPN | 16 | -0.54 | 0.6522 | 1.03 | 0.3020 |

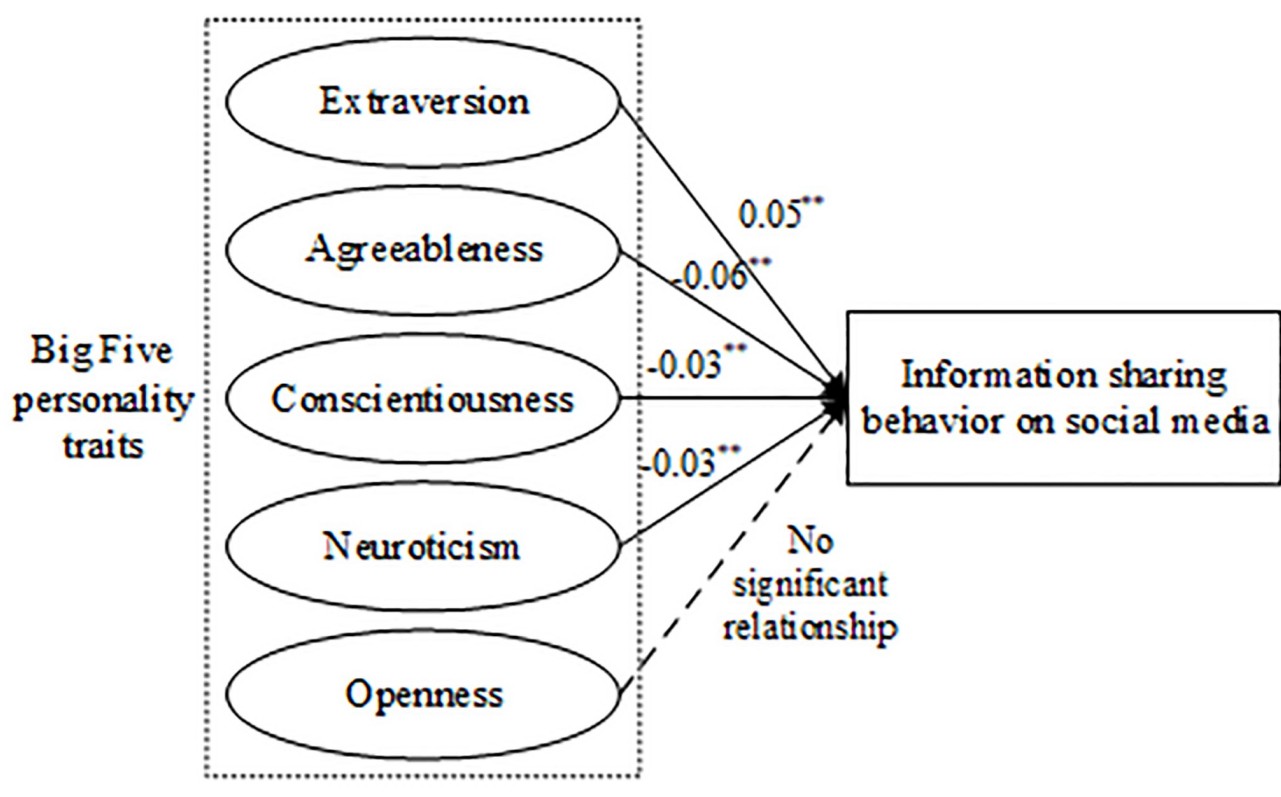

**Fig 10. Revised theoretical model.**

on social media, while agreeableness, conscientiousness, and neuroticism negatively correlate with it. In future studies, it will be important to investigate these personality traits more extensively.

## Supporting information

**S1 Checklist. PRISMA 2020 checklist.**
(PDF)

**S1 Data.**
(DTA)

## Author Contributions

**Conceptualization:** Hao Lin.

**Data curation:** Hao Lin, Chundong Wang, Yongjie Sun.

**Investigation:** Hao Lin, Chundong Wang, Yongjie Sun.

**Methodology:** Hao Lin.

**Project administration:** Chundong Wang.

**Software:** Hao Lin.

**Supervision:** Chundong Wang.

**Validation:** Hao Lin.

**Visualization:** Hao Lin.

**Writing – original draft:** Hao Lin.

**Writing – review & editing:** Hao Lin, Chundong Wang, Yongjie Sun.

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
