## [Decision Letter · Decision Letter 0]

14 Feb 2024

PONE-D-23-41029How Big Five Personality Traits Influence Information Sharing on Social Media: A Meta AnalysisPLOS ONE

Dear Dr. Sun,

Thank you for submitting your manuscript to PLOS ONE. After careful consideration, we feel that it has merit but does not fully meet PLOS ONE’s publication criteria as it currently stands. Therefore, we invite you to submit a revised version of the manuscript that addresses the points raised during the review process.

Reviewers' comments must be addressed thoroughly with equal attention. They should be considered as complementary in the implementation of revisions. Please note that Reference 58 appears to be subject to an Expression of Concern relating to the data reported in that article. Please consider finding an alternative source to support the relevant statements, or if such a source is not available, please remove or revise the statements supported by this reference.

We look forward to receiving your revised manuscript.

Kind regards,

Simone Varrasi

Academic Editor

PLOS ONE

“This work was supported by National Natural Science Foundation of China-Joint Fund Project [U1536122], Key Special Project of “Technology Boosts Economy 2020” by Ministry of Science and Technology [SQ2020YFF0413781], and Pilot Demonstration Project of Big Data Industry Development [Big data intelligent analysis and service platform for language barrier rehabilitation applications].”

“This work was supported by National Natural Science Foundation of China-Joint Fund Project [U1536122], Key Special Project of “Technology Boosts Economy 2020” by Ministry of Science and Technology [SQ2020YFF0413781], and Pilot Demonstration Project of Big Data Industry Development [Big data intelligent analysis and service platform for language barrier rehabilitation applications].

“This work was supported by National Natural Science Foundation of China-Joint Fund Project [U1536122], Key Special Project of “Technology Boosts Economy 2020” by Ministry of Science and Technology [SQ2020YFF0413781], and Pilot Demonstration Project of Big Data Industry Development [Big data intelligent analysis and service platform for language barrier rehabilitation applications].”

5. PLOS requires an ORCID iD for the corresponding author in Editorial Manager on papers submitted after December 6th, 2016. Please ensure that you have an ORCID iD and that it is validated in Editorial Manager. To do this, go to ‘Update my Information’ (in the upper left-hand corner of the main menu), and click on the Fetch/Validate link next to the ORCID field. This will take you to the ORCID site and allow you to create a new iD or authenticate a pre-existing iD in Editorial Manager. Please see the following video for instructions on linking an ORCID iD to your Editorial Manager account: https://www.youtube.com/watch?v=_xcclfuvtxQ.

6. We notice that your supplementary figure and tables are included in the manuscript file. Please remove them and upload them with the file type 'Supporting Information'. Please ensure that each Supporting Information file has a legend listed in the manuscript after the references list.

Additional Editor Comments:

Dear Authors,

many thanks for your submission. Please reply to both the Reviewers and revise the manuscript accordingly, taking thoroughly into account their concerns.

Reviewers' comments:

Reviewer's Responses to Questions

**Comments to the Author**

1. Is the manuscript technically sound, and do the data support the conclusions?

Reviewer #1: Yes

Reviewer #2: Partly

2. Has the statistical analysis been performed appropriately and rigorously? 

Reviewer #1: No

Reviewer #2: Yes

3. Have the authors made all data underlying the findings in their manuscript fully available?

Reviewer #1: Yes

Reviewer #2: Yes

4. Is the manuscript presented in an intelligible fashion and written in standard English?

Reviewer #1: No

Reviewer #2: Yes

5. Review Comments to the Author

Reviewer #1: Dear Authors,

Your work provides cognitively interesting data on personality.

The study is legitimate, but the text is laconic.

It is typical of a Case Study Report, not a Research Article.

Therefore, please improve the editing style appropriate for a Research Article.

Also, please let me know if you used artificial intelligence in your work?

If so, to what extent?

In addition, please refer to the impact of experience on personality in the discussion.

I suggest referring to: https://doi.org/10.3389/fpsyg.2022.854804.

After all, verify the abstract and change the keywords to something other than the title.

Reviewer #2: This is a rather intriguing review, acknowledging the significant role of personality. The reviewers have put forth the following observations:

1. Elucidate the trajectory of your research in the introduction, and underscore why this trajectory is of consequence. Despite the current introduction being quite direct and detailing the structure of this paper, it falls short of a specific delineation of the research problem and background.

2. It is imperative to scrutinise current analogous reviews and accentuate the disparities between this manuscript and them.

3. The discussion warrants a more profound exploration.

4. Kindly delve deeper into the potential implications of the findings of this article.

6. PLOS authors have the option to publish the peer review history of their article (what does this mean?). If published, this will include your full peer review and any attached files.

Reviewer #1: No

Reviewer #2: No

---

## [Author Response · Author response to Decision Letter 0]

7 Apr 2024

For point-to-point responses to reviewer comments, please refer to Response to Reviewerss.pdf.

---

## [Decision Letter · Decision Letter 1]

29 Apr 2024

PONE-D-23-41029R1How Big Five Personality Traits Influence Information Sharing on Social Media: A Meta AnalysisPLOS ONE

Dear Dr. Sun,

Thank you for submitting your manuscript to PLOS ONE. After careful consideration, we feel that it has merit but does not fully meet PLOS ONE’s publication criteria as it currently stands. Therefore, we invite you to submit a revised version of the manuscript that addresses the points raised during the review process.

**Please address the comments made by Reviewers. One of them recommended to reject the manuscript. Is it required to discuss their comments as limitations of you research, and to improve the structure of the article as suggested by both scholars. After these changes, your work will be finally evaluated for its suitability for the Journal.**

We look forward to receiving your revised manuscript.

Kind regards,

Simone Varrasi

Academic Editor

PLOS ONE

Reviewers' comments:

Reviewer's Responses to Questions

**Comments to the Author**

1. If the authors have adequately addressed your comments raised in a previous round of review and you feel that this manuscript is now acceptable for publication, you may indicate that here to bypass the “Comments to the Author” section, enter your conflict of interest statement in the “Confidential to Editor” section, and submit your "Accept" recommendation.

Reviewer #1: All comments have been addressed

Reviewer #2: (No Response)

2. Is the manuscript technically sound, and do the data support the conclusions?

Reviewer #1: Yes

Reviewer #2: No

3. Has the statistical analysis been performed appropriately and rigorously? 

Reviewer #1: Yes

Reviewer #2: No

4. Have the authors made all data underlying the findings in their manuscript fully available?

Reviewer #1: Yes

Reviewer #2: No

5. Is the manuscript presented in an intelligible fashion and written in standard English?

Reviewer #1: Yes

Reviewer #2: No

6. Review Comments to the Author

**Reviewer #1:** Dear Authors,

thank you for inviting me to review your work again and taking into account the suggestions for improvement. Since it is now an original article this excessive number of sections and subsections makes the content difficult to perceive. Therefore, consider rebuilding the content breakdown to a few main sections. Other than that, I have no objections to the content, methodology and pragmatics.

**Reviewer #2: **The topic on which this manuscript focuses is of great interest. However, the manuscript currently has the following problems: (1) the selected database is not representative, (2) the structure and presentation of the manuscript are not clear enough, (3) the manuscript does not fully discuss the issues of concern, and does not combine the analysis results There is also no focus on cultural heterogeneity across regions. In summary, the reviewers believe that the manuscript does not meet the journal's requirements.

7. PLOS authors have the option to publish the peer review history of their article (what does this mean?). If published, this will include your full peer review and any attached files.

Reviewer #1: No

Reviewer #2: No

---

## [Author Response · Author response to Decision Letter 1]

30 Apr 2024

We submit a 'Response to Reviewers.pdf'.

---

## [Editor Report · Decision Letter 2]

1 May 2024

How Big Five Personality Traits Influence Information Sharing on Social Media: A Meta Analysis

PONE-D-23-41029R2

Dear Dr. Sun,

We’re pleased to inform you that your manuscript has been judged scientifically suitable for publication and will be formally accepted for publication once it meets all outstanding technical requirements.

Kind regards,

Simone Varrasi

Academic Editor

PLOS ONE
---

## [Editor Report · Acceptance letter]

15 May 2024

PONE-D-23-41029R2 

PLOS ONE

Dear Dr. Sun, 

I'm pleased to inform you that your manuscript has been deemed suitable for publication in PLOS ONE. Congratulations! Your manuscript is now being handed over to our production team.

Kind regards, 

on behalf of

Dr. Simone Varrasi 

Academic Editor

PLOS ONE